# Ballistic conductance with and without disorder in a boundary-driven XXZ spin chain

**Adam J. McRoberts**[1,2] **and Roderich Moessner**[2]

[1] International Centre for Theoretical Physics,
Strada Costiera 11, 34151, Trieste, Italy
[2] Max Planck Institute for the Physics of Complex Systems,
Nöthnitzer Str. 38, 01187 Dresden, Germany

## Abstract

Motivated by recent experiments on Google's sycamore NISQ platform on the spin transport resulting from a non-unitary periodic boundary drive of an XXZ chain, we study a classical variant thereof by a combination of analytical and numerical means. We find the classical model reproduces the quantum results in remarkable detail, and provides an analytical handle on the nature and shape of the spin transport's three distinct regimes: ballistic (easy-plane), subdiffusive (isotropic) and insulating (easy-axis). Further, we show that this phenomenology is remarkably robust to the inclusion of bond disorder – albeit that the transient dynamics approaching the steady states differs qualitatively between the clean and disordered cases – providing an accessible instance of ballistic transport in a disordered setting.

# 1   Introduction

The Heisenberg chain and its descendants, especially the XXZ model, have long served as canonical models of magnetism and played a tremendously important role in the understanding of low-dimensional systems [1–5]. In particular, the quantum $S = \frac{1}{2}$ XXZ chain is a quintessential example of the connection between anomalous equilibrium hydrodynamics and integrability [6–25]; and even non-integrable spin chains can evince anomalous dynamics for remarkably long timescales [26–32].

   Beyond the (near-)equilibrium paradigm, recent years have seen an increased interest in far-from-equilibrium phenomena [33–52] and the physics of open [53–69] and driven (Floquet) systems [70–86], where the transport phenomenology can differ markedly from the linear response regime.

   In this vein, recent experimental work on Google's noisy intermediate-scale quantum (NISQ) platform [87] has investigated the effect of edge driving on the $S = \frac{1}{2}$ XXZ chain, and yielded a detailed – in particular, fully spatially-resolved – spin transport phenomenology on a chain of $L = 26$ sites, comprising non-equilibrium steady states (NESS) of ballistic, subdiffusive, and insulating nature (for easy-plane, isotropic, and easy-axis anisotropy, respectively), backed up by numerics as well as prior work on NESS [33].

   Another paradigmatic approach to altering the transport and hydrodynamics of a given system is the introduction of disorder. This often leads to parametrically slower dynamics compared to the clean system – turning, say, diffusion into subdiffusion [88–91] or (many-body) localisation [92–96]. However, the asymptotic behaviour of disordered models is often particularly difficult to sleuth out: analytic solutions are usually unavailable, numerics on quantum systems are typically limited to small sizes, and the outsized effects of rare (Griffiths) regions [97,98] and long crossovers [99–106] can obscure the true infinite-time dynamics.

   Here, we report a study of a boundary-driven *classical* XXZ chain – closely analogous to the Google experiment [87] – considering both clean and bond-disordered versions of this system.

   We obtain a detailed and largely analytic set of results, supported by numerical simulations. Our main findings are the following: first, we show that the NESS for this set-up can be, relatively straightforwardly, explicitly derived; this, in particular, yields the anomalous exponent for the isotropic case $\Delta = 1$. And second, surprisingly, these steady-states and all of their associated transport phenomenology agree in considerable detail with the experimental results on the *quantum* system [87], providing a rich and notable non-equilibrium instance of a quantum-classical correspondence.

   Further, we show that the phenomenology of the clean system – in particular, the three transport regimes (ballistic, subdiffusive, and insulating) – is remarkably robust to the addition of bond disorder. This model thus provides an analytically tractable instance of *ballistic* transport in a disordered setting, in the sense that the (distribution of) currents through the NESS – and thus the conductance of the chain – is independent of the system size. However,

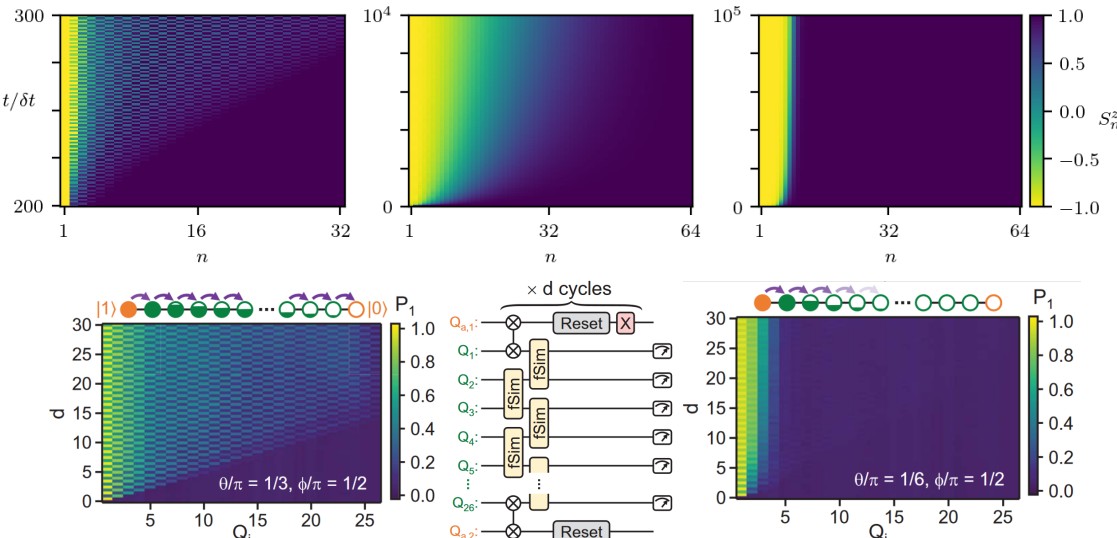

Figure 1: Overview of the different dynamical regimes, and comparison to the quantum experiments. The figures in the second row are copied, for ease of comparison, from Ref. [87]. From left to right: easy-plane ($\Delta = 0.6$), isotropic ($\Delta = 1$), and easy-axis ($\Delta = 1.5$). Value of $S^z(n, t)$ is shown in each case. The easy-plane features a front which spreads ballistically across the chain. There is no transport across the chain in the easy-axis case: a domain wall (which is stable under these dynamics) is formed. In the isotropic case, the spin excitations can only move subdiffusively, but will eventually reach the other side.

the relaxation timescales for reaching the steady state are diffusive in the sense that they scale quadratically in the system size. The transport regimes of the clean model only break down at strong disorder, where weak bonds can decompose the chain into disconnected segments, and a tunable subdiffusive exponent appears, in analogy to Ref. [106].

In the following, we define the model and derive the aforementioned results in turn.

## 2 Model

We consider a classical chain of unit-length spins $S_i \in S^2$ with boundary-driven Floquet dynamics. A single time-step $t \mapsto t + \delta t$ consists of three parts. Between times $t$ and $t + \delta t/2$, odd bonds $(1, 2), (3, 4), ..., (L-1, L)$ evolve, followed by the even bonds between times $t + \delta t/2$ and $t + \delta t$. Finally, the boundary spins are reset, $S_1 = -\hat{z}$, $S_L = +\hat{z}$ (cf. iSWAP($\pi/2$) gates of the quantum experiment [87]).

The bond dynamics is generated by the restriction of the XXZ Hamiltonian to the pair of sites being evolved,

$$\mathcal{H} = -\sum_{i=1}^{L-1} J_i \left( S_i^x S_{i+1}^x + S_i^y S_{i+1}^y + \Delta S_i^z S_{i+1}^z \right) . \tag{1}$$

The anisotropy $\Delta$ is the principal tuning parameter. Below, we first consider a clean system, $J_i = 1$, followed by the bond disordered case.

Now, in the bulk of the system, the $z$-magnetisation is locally conserved, and the spin current across bond $i$ over the time-step $t \mapsto t + \delta t$ is given by

$$\begin{cases} j_i(t) = -S_i^z(t + \delta t/2) + S_i^z(t) & i \text{ odd} \\ j_i(t) = -S_i^z(t + \delta t) + S_i^z(t + \delta t/2) & i \text{ even.} \end{cases} \tag{2}$$

Crucially, however, magnetisation can flow into and out of the system at the boundaries, at the moment when the boundary spins are reset.

Now, experiments on the quantum $S = \frac{1}{2}$ version of this model [87] prepared the initial state

$$|\psi\rangle = |\downarrow\uparrow\uparrow \dots \uparrow\uparrow\rangle, \tag{3}$$

which, since $|\downarrow\uparrow\rangle$ is not an eigenstate of the two-site $S = \frac{1}{2}$ XXZ Hamiltonian, evolves non-trivially with time. Classically, however, perfectly anti-parallel spins do not evolve, requiring a small initial perturbation to seed the dynamics. In our simulations, we tilt the second spin in the XZ plane, and use the initial state

$$\begin{aligned} S_1 &= -\hat{z}, \ \ S_2 = \chi\hat{x} + \sqrt{1 - \chi^2}\hat{z}, \\ S_3 &= \dots = S_L = +\hat{z}, \end{aligned} \tag{4}$$

where, in what follows, we set $\chi = 0.01$. This leads to a delay in the onset of the dynamics, and the injected current rises exponentially up to a time $t \approx 10^2\delta t$, Fig. 3(a). After this delay, the dynamics proceeds essentially in the same manner as in the quantum case, and, in particular, we show that the asymptotics are fundamentally equivalent.

## 3 Clean spin transport

Fig. 1 shows the numerically obtained spatiotemporal behaviour of $S^z$ for different $\Delta$, and a comparison with the quantum case: ballistic transfer of spin between the boundaries for $\Delta < 1$ (easy-plane), and no spin transport because of the formation of a domain wall for $\Delta > 1$ (easy-axis). At the isotropic point ($\Delta = 1$), the spin transport is subdiffusive. In the following, we present an analytical understanding by deriving these non-equilibrium steady states (NESS).

### 3.1 Non-equilibrium steady states

At late times, under the Floquet dynamics of Eq. (1), we approach a non-equilibrium steady-state (NESS) which satisfies the stroboscopic condition

$$S_i(t) = S_i(t + \delta t), \quad \forall i. \tag{5}$$

We consider first the limit of small time-steps $\delta t \to 0$ (or large driving frequency $\omega \to \infty$). This limit is well-defined, and corresponds to the continuous-time dynamics of the XXZ Hamiltonian (1), subject to the boundary conditions $S_1 = -\hat{z}$, $S_L = +\hat{z}$. The steady-state condition becomes

$$\dot{S}_i(t) = 0, \quad \forall i, \quad \omega \to \infty. \tag{6}$$

These equations may be solved to arbitrary precision. Explicitly, the $z$-components $S_i^z = z_i$ satisfy

$$\frac{z_i}{\sqrt{1 - z_i^2}} = \Delta \frac{J_i z_{i+1} + J_{i-1} z_{i-1}}{J_i \sqrt{1 - z_{i+1}^2} + J_{i-1}\sqrt{1 - z_{i-1}^2}}, \tag{7}$$

with $z_1 = -1$, $z_L = +1$, and the in-plane components have a constant azimuthal angle.

We note that the steady-states obtained from Eq. (7) strongly resemble the steady-states of the quantum model in a similar limit [33], though we point out that there is no *a priori*

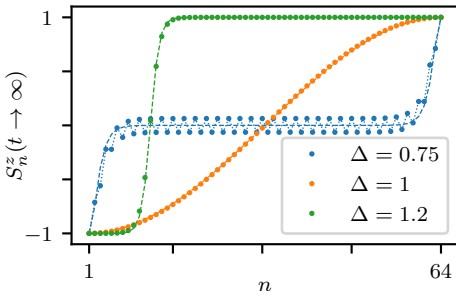

Figure 2: Comparison of the NESS obtained from the finite-frequency simulations ($t = 10^5 \delta t$, $\delta t = \pi/2$, dots and dotted line) and the infinite-frequency consistency equations (7) (dashed lines). The spatial oscillations in the easy-plane case are a finite-frequency effect.

reason for this correspondence, particularly since the $\omega \to \infty$ limit of the quantum model is integrable, and the classical model is not. We show an example steady-state for each of the dynamical regimes in Fig. 2.

## 3.2 Transport regimes

The spin current in the steady-state determines the asymptotic transport regime. Strictly speaking, of course, the infinite-frequency NESS has no currents (it is defined by every spin being static), and the finite-frequency NESS is defined by the stroboscopic condition (5) and is, thus, in general, a different state. In the high-frequency limit, however, we approximate the finite-frequency NESS with the infinite-frequency NESS, but we apply the finite-frequency dynamics to obtain the asymptotic, steady-state values of the current $j_\infty(L) = j(t \to \infty; L)$.

Since we are considering a boundary-drive, i.e., there is no driving in the bulk, any steady-state current is proportional to the conductance $G(L) \sim L^{1-1/\alpha}$, where $\alpha$ is the dynamical exponent of the hydrodynamic scaling relation $x \sim t^\alpha$. And since any steady-state current must be uniform, it suffices to consider the current on the final bond, for which we have

$$G(L) \sim j_\infty(L) \sim 1 - z_{L-1} \tag{8}$$

(see App. A).

It remains only to calculate the steady-states to find the conductances and corresponding transport exponents. In the easy-plane case we find that $z_{L-1} = \Delta$ is length-independent; the exact solution at the isotropic point is $z_{L-1} = \cos(\pi/L)$; and in the easy-axis case we find $1 - z_{L-1} \sim e^{-L}$ (we discuss these solutions in more detail in App. B). That is, we have the conductances, and corresponding dynamical exponents,

$$\begin{cases} G \sim 1 - \Delta & \Rightarrow & \alpha = 1 & \Delta < 1 \text{ (ballistic)} \\ G \sim L^{-2} & \Rightarrow & \alpha = 1/3 & \Delta = 1 \text{ (subdiffusive)} \\ G \sim e^{-L} & \Rightarrow & \alpha = 0 & \Delta > 1 \text{ (localised)}. \end{cases} \tag{9}$$

The appellation of the regimes – ballistic, subdiffusive, localised – follow from the identification of the scaling behaviour of the corresponding conductances in the case of 'standard' transport phenomena in response to the establishment of a potential difference across a chain.

## 3.3 Finite-frequency simulations

We verify the predictions of the infinite-frequency calculation by performing numerical simulations of the dynamics at finite frequency. Starting from the initial state (4), we find the fol-

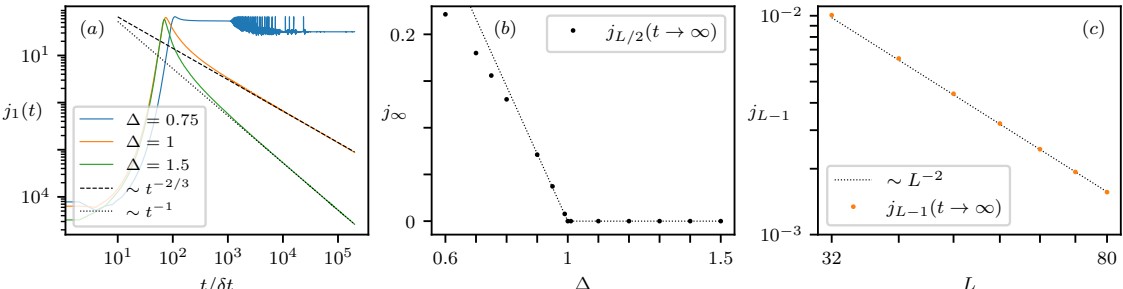

Figure 3: Non-equilibrium spin transport in the clean model. (a) Time-dependence of the injected current $j_1(t)$ in the three different regimes: there is no decay, corresponding to ballistic transport, in the easy plane; $t^{-2/3}$ decay at the isotropic point implies subdiffusion; and $t^{-1}$ decay corresponds to localisation in the easy-axis case. The initial exponential increase up to $t \approx 10^2 \delta t$ is a transient effect due to the nature of the initial state (4). (b) Steady-state spin current as a function of $\Delta$; only the easy-plane ($\Delta < 1$) supports an $L$-independent current. The dotted line shows the linear dependence on $1 - \Delta$ near the isotropic point. (c) Steady-state spin current as a function of $L$ at the isotropic point, which decays $\sim L^{-2}$ as predicted by the infinite-frequency analysis (9).

lowing (cf. Fig. 1): for easy-plane anisotropy ($\Delta < 1$), a ballistic magnetisation front crosses the chain, and the NESS is attained at long times – the spin profile symmetrises between the $-\hat{z}$ and $+\hat{z}$ boundaries, oscillates around $S^z = 0$ near the middle of the chain, and a spatially-uniform, non-zero spin current flows through the bulk.

In the easy-axis ($\Delta > 1$), a sharp domain wall quickly develops near the left-boundary, and the injection of spin into the chain effectively halts. The true NESS is symmetric about the centre of the chain, but this state will only be attained after some exponential timescale (in the system size), as, once the domain wall is a few sites away from the boundary, the system is in a very good approximation of a steady state (as measured by Eqs. (7)).

The isotropic point ($\Delta = 1$) is the transition between these ballistic and localised regimes, and the magnetisation profile builds up subdiffusively.

The dynamical exponent can be measured at finite time by the injected current $j_1(t)$. This can be understood from the following hydrodynamic argument: the boundary drive injects magnetisation through the edge of the system at a constant rate, except that this will be slowed if there is a build-up of spin near the boundary. Since the magnetisation is not driven across in the bulk, the scaling relation implies that, by time $t$, it will spread a distance $x \sim t^\alpha$ into the chain. That is, it moves away from the edge at an effective rate $t^\alpha/t$, which is precisely the current that can be injected at time $t$, i.e., $j_1(t) \sim t^{\alpha-1}$.

We plot the injected current for three representative anisotropies in Fig. 3(a). As noted in the discussion of the initial state, there is an initial exponential increase (from $j_1(0) \sim \chi^2$) as the system moves away from the unstable stationary state ($-\hat{z}, +\hat{z}, +\hat{z}, ...$), but after this delay we find the dynamical regimes predicted by the infinite-frequency calculation,

$$\begin{cases} j_1(t) \sim \text{const.} & \Delta < 1 \\ j_1(t) \sim t^{-2/3} & \Delta = 1 \\ j_1(t) \sim t^{-1} & \Delta > 1. \end{cases} \tag{10}$$

We also verify explicitly in Fig. 3(c) that the conductance at the isotropic point scales as $G(L) \sim L^{-2}$.

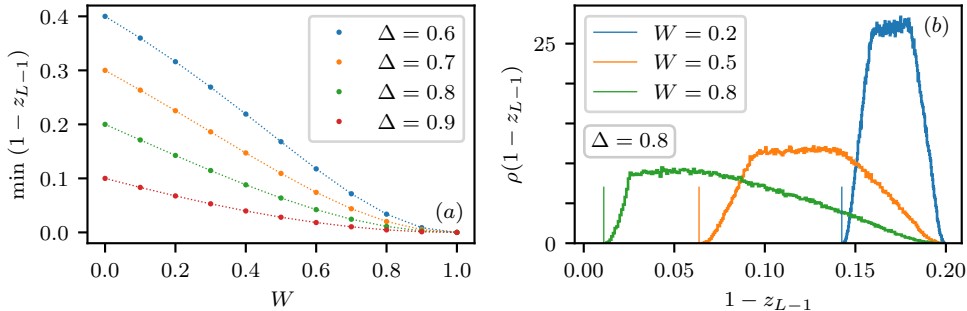

Figure 4: Infinite-frequency analysis of the disordered transport. (a) Conjectured minimum value (as a function of disorder realisation) of $1-z_{L-1} \sim G(L)$ as a function of disorder strength $W$ for various easy-plane anisotropies. (b) Probability density of $1-z_{L-1}$ for various disorder strengths at $\Delta = 0.8$, obtained by numerically solving the consistency equations (7) for $3 \times 10^5$ independent realisations of disorder. The vertical lines denote the (conjectured) minimum possible value, cf. (a), with no sets of random couplings found to violate the bound.

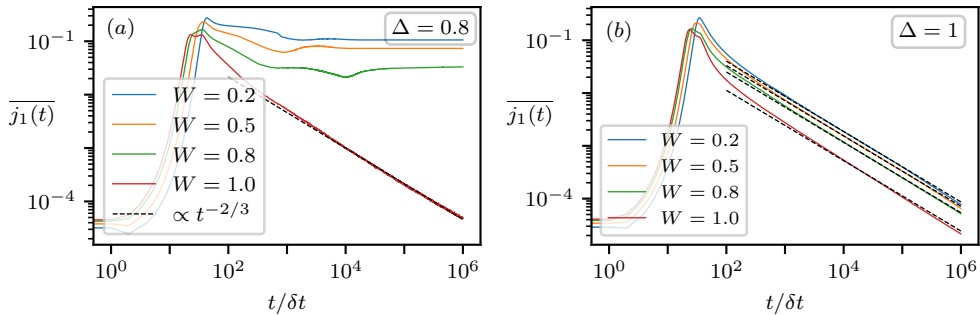

Figure 5: Non-equilibrium spin transport in the disordered model. (a) Time-dependence of the disorder-averaged injected current $\overline{j_1(t)}$ for $\Delta = 0.8$ and selected disorder strengths $W$; there is no decay, corresponding to ballistic transport, for $W < 1$. The current decays subdiffusively ($\alpha = 1/3$) for $W = 1$. (b) Decay of $\overline{j_1(t)}$ at the isotropic point $\Delta = 1$, where subdiffusive decay is observed for all disorder strengths.

## 3.4 Classical-quantum correspondence

The classical model we have studied here exhibits a striking similarity to the quantum experiments [87]; indeed, the only real difference in the observed phenomena is a transient, initial time lag due to the nature of the initial state (4) (cf. Fig. 3(a)).

The spatio-temporal pictures (Fig. 1) are qualitatively very similar: in both the classical and quantum cases the ballistic front in the easy-plane is clearly visible; in the easy-axis, the domain wall is also easily observed in both pictures.

More concretely, the transport regimes, measured by the decay of the injected current $j_1(t)$, are precisely equivalent in both the classical model and the quantum experiments, and are in accordance with the predictions of the infinite-frequency analysis of the steady-states in §3.1.

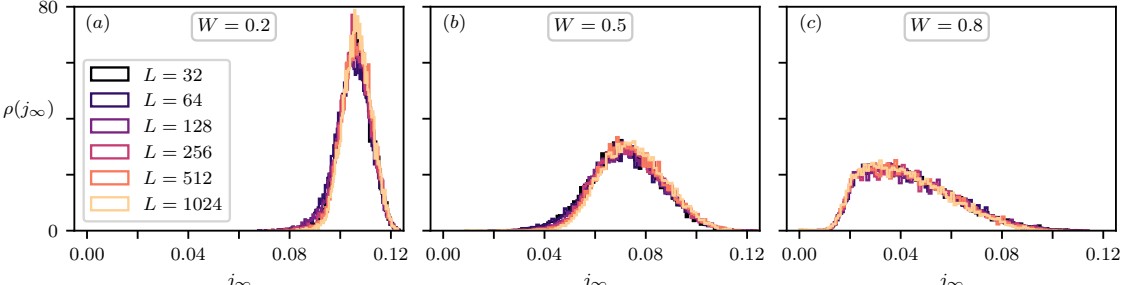

Figure 6: Ballistic conductance in the disordered chain. Histograms of the distribution of steady-state currents, $\rho(j_\infty(L))$ for $\Delta = 0.8$ and $W = 0.2$ (a), $W = 0.5$ (b), and $W = 0.8$ (c). Each histogram is constructed from the final values $j_{L-1}(t_f)$ for $t_f = 10^6 \delta t$, $\delta t = \pi/2$, over $10^4$ realisations of disorder. We find that $\rho(j_\infty)$ is independent of system size, at least up to $L = 1024$, and that, as predicted by the infinite-frequency analysis, the steady-state current distribution has a nonzero lower bound. Note that for $W = 0.8$, due to the increased relaxation times (cf. Fig 7), we have, only for this figure, started from the initial state $S_2 = ... = S_{L-1} = \hat{x}$ in order that the steady states are reached before the end of the simulation.

## 4 Disordered spin transport

Having shown that the clean classical model has the same transport regimes as the quantum model [87], we next investigate how the phenomenology is affected by the introduction of random couplings. We will primarily consider the easy-plane regime, and, in §4.4, the isotropic point. The easy-axis case is not particularly interesting: there is no transport in the clean case, and, unsurprisingly, there is also no transport in the disordered case.

Now, the fact that the ballistic and subdiffusive regimes appear in a real experiment implies at least perturbative stability on the length-scales probed; in the following, we show that this phenomenology is remarkably robust, with even ballistic conductance persisting in the presence of disorder. We will see, however, that the approach to the steady state is qualitatively affected by the introduction of random couplings.

Concretely, we study the model (1) with couplings $J_i$ drawn independently from the uniform distribution over the interval $[1 - W, 1]$, where $W$ is the disorder strength. We keep the two outer bonds fixed at $J_1 = J_{L-1} = 1$, as these provide the connection to the external reservoirs. We use the same initial state (4) as in the clean model.

### 4.1 Transport observables

We will continue to characterise the spin transport using the steady-state current $j_\infty(L)$, and, at finite-time, the injected current $j_1(t)$. The introduction of disorder, however, raises two important points: (i) the steady-state current will depend on the particular sample of disordered couplings, and (ii) the timescales on which the steady-state is attained may become, in practice, inaccessible.

We deal with (i) by considering the full probability distribution of steady-state currents $\rho(j_\infty)$ and not just the disorder-average $\overline{j_\infty}$. We will return to (ii) when we perform finite-frequency simulations in §4.3.

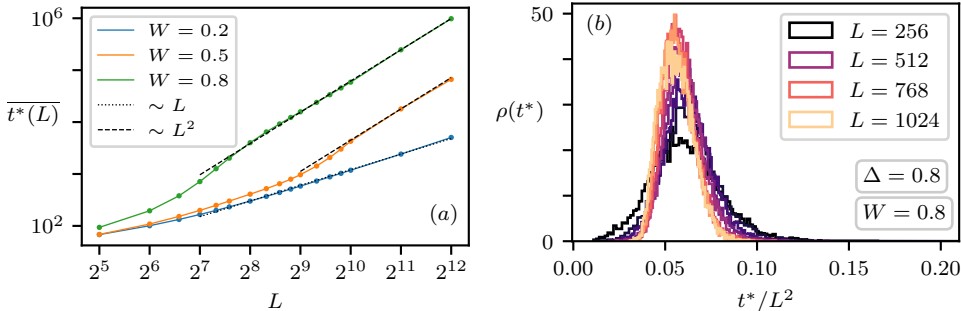

Figure 7: (a) Disorder-averaged crossing times $\overline{t^*}$ as a function of system-size for selected disorder strengths – we find that, whilst we expect $t^* \sim L^2$ in the limit of large systems, we can observe superdiffusive, or even ballistic scaling $t^* \sim L$ up to chain lengths $L \approx 10^3$. (b) Distribution of crossing times $\rho(t^*)$ for various system sizes for $\Delta = 0.8$ and $W = 0.8$. We find that a diffusive scaling collapse describes the data reasonably well.

## 4.2   Stability of the ballistic regime

Again, we begin by considering the steady-states in the infinite-frequency limit, given by the consistency equations (7), but now with random couplings. We show that, in the easy-plane, so long as there is some lower bound to the coupling strengths (i.e., $W < 1 \Rightarrow J_i > 0$), the ballistic regime survives in the sense that $G(L) \sim j_\infty \sim O(L^0)$. This stability derives from the fact that the disordered consistency equations are fundamentally limited in the extent to which they can modify the NESS of the clean model (We discuss typical disordered steady states in App. C).

To establish that the ballistic regime survives, we need to show that there remains a finite, length-independent difference between $z_{L-1}$ and $z_L = 1$, since, cf. Eq. (8), this yields a finite conductance $G(L) \sim 1 - z_{L-1}$.

We conjecture that the minimum conductance $G(L)$ (as a function of the disorder realisation) is obtained from the couplings $J_2 = ... = J_{L-2} = 1-W$, and that this, indeed, corresponds to a finite, length-independent lower bound on $G(L)$. We show this minimum conductance as a function of the disorder strength $W$, for various anisotropies $\Delta < 1$, in Fig. 4(a).

We do not have a proof that this is the minimum. However, we show histograms of the probability density of $1 - z_{L-1}$ for $\Delta = 0.8$ and selected $W$ in Fig. 4(b), lending numerical credence to our assertion that the conductance $G(L) \sim 1 - z_{L-1}$ has a finite lower bound in the easy-plane – none of the $3 \times 10^5$ randomly drawn sets of couplings are found to violate the conjectured bound.

## 4.3   Finite-frequency simulations with disorder

We again verify the predictions of the infinite-frequency analysis with finite-frequency simulations. For each $W$ and $\Delta$ considered, we evolve over $10^4$ independent realisations of disorder.

We show in Fig. 5(a) that the disorder-averaged injected current $\overline{j_1(t)}$ continues to show ballistic behaviour at finite time: for $W < 1$, $\overline{j_1(t)}$ approaches a constant, finite value, and does not decay. This is the same ballistic behaviour as seen in the clean model, cf. Fig. 3(a).

Further, we show the steady-state current distribution $\rho(j_\infty(L))$ for $\Delta = 0.8$ and $W = 0.2$, 0.5 and 0.8 in Fig. 6, and find the ballistic behaviour predicted by the infinite-frequency analysis. That is, the distribution is length-independent – at least up to the system sizes of $L = 1024$ for which we can attain the NESS in our simulations – and bounded below by some finite value.

We should address, however, the question of the timescale on which the length-independent steady-state distribution is set up. Indeed, this timescale need not (and, we will show, does not) have the same length-scaling as the steady-state current: the arguments of §4.2 imply that the steady-state current is primarily determined by the bonds near the edge of the chain; but to attain the NESS from the initial state (4) the magnetisation must propagate across the full length of the chain, which clearly involves every bond.

To account for this, we define a crossing time $t^*$ as the first time for which the *extracted* current $j_{L-1}(t^*) > 10^{-4}$ – though the exact threshold, is, of course, arbitrary – and determine how this time scales with the system size.

It is here that the disordered nature of the chain becomes visible – the distribution of crossing times $t^*$ exhibits diffusive, not ballistic, behaviour, on the longest length-scales we can simulate. We show in Fig. 7 that the crossing time scales as $t^* \sim L^2$ (i.e., diffusively); though we show in Fig. 7(a) that it can appear to be superdiffusive or even ballistic up to surprisingly large system sizes $L \gtrsim 10^3$ (depending on $W$), which would be relevant to experiments on near-term quantum devices.

## 4.4 Subdiffusion at strong disorder

Finally, let us briefly address what happens when the disorder strength reaches $W = 1$. The arguments of §4.2 no longer apply, and, as shown in Fig. 5(a), we find a subdiffusive scaling of the disorder average of the injected current, $\overline{j_1(t)} \sim t^{-2/3}$. We discuss a possible explanation for this observation in App. D.

We note that this is the same subdiffusive exponent ($\alpha = 1/3$) as observed at the isotropic point in the clean case, and we show in Fig. 5(b) that, in fact, this subdiffusive exponent is observed for all disorder strengths at the isotropic point.

## 5 Conclusions

In this work, we have provided a comprehensive analysis of the boundary driven classical XXZ chain. We have first shown that the classical chain reproduces the phenomenology of the quantum experiments in exquisite detail [87], and provided an analytical understanding of the transport regimes in terms of the non-equilibrium steady states.

We have further shown that the ballistic regime in the easy-plane survives the introduction of bond disorder, in the sense that the conductance in the steady state is independent of the system size, approaching some limiting distribution (cf. Fig. 6). And, whilst the timescale on which the NESS is attained grows diffusively at the longest system sizes, even this measure of the transport can appear to be ballistic for surprisingly long chains. Moreover, we have provided an analytic understanding of the remarkable stability of the ballistic conductance in terms of the infinite-frequency steady states.

This work immediately raises two further research directions. Firstly, how generally, and under which conditions, does there appear such a detailed agreement between small-spin quantum dynamics on one hand, and classical dynamics on the other? And, secondly, how far-reaching is the robustness against the addition of disorder of varying types and strengths – in particular, is this ballistic conductance in the presence of disorder also observed in the quantum case? We note that the latter items lend themselves to immediate implementation on the same platform as the original experiment of Ref. [87].

# Acknowledgements

We thank Dima Abanin, Pieter Claeys, Ferdinand Evers, Sarang Gopalakrishnan, David Huse and Xiao Mi for many useful discussions.

**Funding information**    This work was in part supported by the Deutsche Forschungsgemeinschaft under grants Research Unit FOR 5522 (project-id 499180199) and the cluster of excellence ct.qmat (EXC 2147, project-id 390858490).

# A  Finite-frequency dynamics

In this appendix we derive the relationship between the values of the spins in the steady-state and the conductance of the chain. We first note that any steady-state current must be spatially uniform, so it suffices to consider only the final two spins $\boldsymbol{S}_{L-1}$ and $\boldsymbol{S}_L$. Now, at the start of a given timestep, $\boldsymbol{S}_L(t) = +\hat{\boldsymbol{z}}$, and, at the end of the timestep $t + \delta t$, $\boldsymbol{S}_L$ is reset to $+\hat{\boldsymbol{z}}$. The magnetisation extracted through the boundary is, therefore,

$$j_{L-1}(t) = 1 - S_L^z(t + \delta t/2) \tag{A.1}$$

(where we have assumed that $L$ is even such that the bond $(L-1, L)$ evolves over the first half of the timestep, but this detail is not important). At the isotropic point $\Delta = 1$, the dynamics of a single bond is particularly straightforward: both spins precess around the total conserved value $\boldsymbol{S}_{L-1} + \boldsymbol{S}_L$. Denoting $\boldsymbol{S}_{L-1}(t)$ by $z_{L-1}$, one thus obtains

$$j_{L-1} = \frac{1}{2}(1 - z_{L-1})(1 - \cos(\delta t/2)). \tag{A.2}$$

By assumption, this is the steady-state current, and thus $j_\infty(L) \sim G(L) \sim 1 - z_{L-1}$. We note that this calculation does not depend on whether we are using the infinite-frequency NESS, or the finite-frequency NESS satisfying the stroboscopic condition (5). The proportionality, $j_\infty(L) \sim 1 - z_{L-1}$, continues to hold away from the isotropic point, cf. Fig. 3(b).

# B  Solution of the consistency equations

We next discuss the solutions of the consistency equations (7), which yield the infinite-frequency steady states. Numerically, the equations can be solved to arbitrary precision (in both the clean and disordered cases), as described in the supplementary of Ref. [28].

In the clean case, we can discuss the solutions in more detail. In the easy-plane case, as stated in the main text, we find $z_{L-1} \sim \Delta$, $L \to \infty$, which implies a length-independent steady-state current and ballistic spin transport. At the isotropic point, the solution can be written down exactly,

$$z_n = \sin\left(\frac{\pi}{L}\left(n - \frac{L+1}{2}\right)\right), \tag{B.1}$$

which is easily verified by substituting it back into the consistency equations (7). We thus obtain

$$z_{L-1}(L) = 1 - \frac{\pi^2}{8}L^{-2} + \frac{\pi^4}{384}L^{-4} + \dots, \tag{B.2}$$

and therefore

$$G(L, \Delta = 1) \sim 1 - z_{L-1} \sim L^{-2}. \tag{B.3}$$

Finally, in the easy-axis case, the solution is given asymptotically by the topological soliton

$$z_n \sim \tanh\left[\left(n - \frac{L+1}{2}\right)\mathrm{arccosh}(\Delta)\right], \quad L \to \infty \tag{B.4}$$

(this solution is derived in the supplementary of Ref. [52]). Only exponentially small deviations from this state are necessary to fulfil the boundary conditions at finite size, and the conductance $G(L) \sim e^{-L}$ is exponentially suppressed.

## C  Typical disordered steady-states

In this short appendix we discuss the typical steady states of the bond disordered systems, shown in Fig. 8. As discussed in the main text, the stability of the ballistic conductance in the easy-plane case is derived from the lower bound on $1 - z_{L-1}$; that is, the fact that the disorder is limited in the extent to which it can modify the steady states. We see in Fig. 8 that the entire steady state is only weakly modified by the introduction of bond disorder.

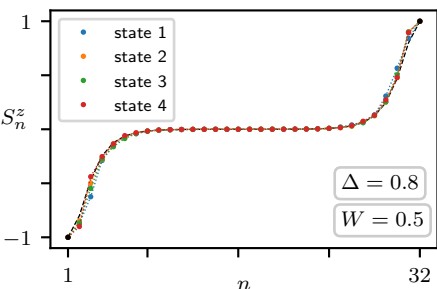

Figure 8: Typical disordered steady states (in the $\omega \to \infty$ limit) for the easy-plane ($\Delta = 0.8$), shown for $W = 0.5$ and four different realisations of disorder. The black dashed line is the clean steady state. This demonstrates graphically that the stability of the ballistic conductance derives from the fact that the steady states are only slightly modified by the introduction of disorder.

## D  Subdiffusion at strong disorder

We turn our attention now to the transition away from the ballistic regime at strong disorder: the arguments of §4.2 do not apply if there is no nonzero lower bound to the coupling strengths, $J_i \geq J_{\min} > 0$, and, as shown in Fig. 5(a), the exponent jumps from ballistic ($\alpha = 1$) to subdiffusive ($\alpha = 1/3$) behaviour at $W = 1$.

However, the explanation does not appear to be so simple as noting that there is no longer any minimum difference between $z_{L-1}$ and $z_L = 1$ in the steady state. Indeed, simply sampling a few realisations of disorder and inspecting the infinite-frequency NESS indicates that there is no sudden change in the typical steady states (since, typically, the bonds near the edge of the chain will not be vanishingly small).

The previous analysis, however, has neglected the fact that, for any NESS, a very small $J_i$ in the bulk limits the total current. For $W < 1$, we could reasonably neglect this detail as our primary interest lies in the transport exponent, and the lower bound to $J_i$ implied that each bond could support some minimum current.

To try to gain some insight into the nature of the strongly-disordered transport, let us consider the power-law disorder distributions used in Ref. [106]. That is, each bond $J_i$ is

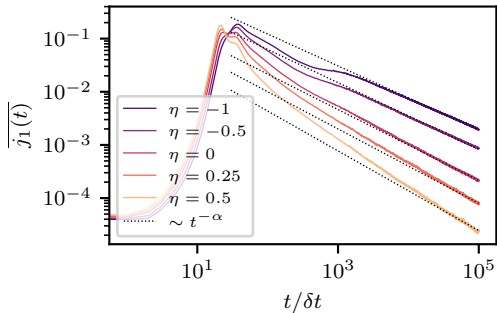

Figure 9: Time-dependence of the disorder-averaged injected current $\overline{j_1(t)}$, for the power-law distributions of disorder, Eq. (D.1). The power-law asymptotes use the subdiffusive exponent $\alpha(\eta)$ predicted by Eq. (D.3). Again, $10^4$ realisations of disorder have been averaged over. $L = 1024$.

drawn from the probability density

$$p(J) = (1-\eta)J^{-\eta}, \quad J \in [0,1], \tag{D.1}$$

for some exponent $\eta \in (-\infty, 1]$. The uniform distribution (with $W = 1$) is recovered for $\eta = 0$.

Now, the expected value of the minimum coupling for a chain of length $L$ scales as

$$\overline{\min_{i=1,\dots,L} J_i} \sim L^{-\frac{1}{1-\eta}}. \tag{D.2}$$

Assuming, since this is a global bottleneck, that this value is directly proportional to the *conductivity* (not the conductance), we obtain

$$G(L) \sim L^{-\frac{1}{1-\eta}-1} \quad \Rightarrow \quad \alpha = \frac{1}{2 + \frac{1}{1-\eta}}, \tag{D.3}$$

which correctly returns $\alpha = 1/3$ at $\eta = 0$ (cf. Fig. 5(a)). We plot the decay of the injected current $\overline{j_1(t)}$ in Fig. 9 for a few different values of $\eta$, and observe reasonable agreement with the subdiffusive exponents predicted by the above scaling analysis.

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
