# Peer review of "Ballistic conductance with and without disorder in a boundary-driven XXZ spin chain"

_SciPost Physics, doi:SciPost Phys. 18, 015 (2025)_

## Round 2 · Referee Report · Anonymous (Referee 1) · 2024-9-22

Strengths

  • timely topic, directly relevant for experiments
  • interesting results, and nice analytical derivation of the non-equilibrium steady state and on the stability of the ballistic transport

Weaknesses

  • some results were already derived in previous papers
  • the quantum-classical correspondence, and especially the role of integrability, is not really explained

Report

As mentioned above, the paper is very fascinating as it shows that the classical Heisenberg chain captures most of the features observed in the boundary driven XXZ chain. The calculations are correct, and the numerical simulations are high quality. The paper deserves publication.

As mentioned above, however, the paper would be of much better quality if the quantum-classical correspondence, and the role of integrability, is better explained. Currently, it mostly looks like a coincidence. We know indeed (for example for the problem of the domain wall of spins or the superdiffusive transport) that there exists some kind of universality class of spin transport given by the classical, integrable, Landau–Lifshitz. The latter can be discretized giving an integrable classical spin chain (also it can be trotterised, see the works in the Prosen group). The authors could run some numerics with the integrable model, showing that the same behavior is observed, or not. This would clarify if the phenomena at hand are actually, also in this case, due to the Landau–Lifshitz equation.

Requested changes

see above.

Recommendation

Publish (meets expectations and criteria for this Journal)

  • validity: high
  • significance: high
  • originality: high
  • clarity: top
  • formatting: perfect
  • grammar: perfect

Author:  Adam McRoberts  on 2024-11-28  [id 5003]

(in reply to Report 1 on 2024-09-22)

Response to the Report of Referee 1

''As mentioned above, the paper is very fascinating as it shows that the classical Heisenberg chain captures most of the features observed in the boundary driven XXZ chain. The calculations are correct, and the numerical simulations are high quality. The paper deserves publication.''

--- We thank the referee for their report, and are pleased that they recommend publication. We address their questions below.

''As mentioned above, however, the paper would be of much better quality if the quantum-classical correspondence, and the role of integrability, is better explained. Currently, it mostly looks like a coincidence. We know indeed (for example for the problem of the domain wall of spins or the superdiffusive transport) that there exists some kind of universality class of spin transport given by the classical, integrable, Landau–Lifshitz. The latter can be discretized giving an integrable classical spin chain (also it can be trotterised, see the works in the Prosen group). The authors could run some numerics with the integrable model, showing that the same behavior is observed, or not. This would clarify if the phenomena at hand are actually, also in this case, due to the Landau–Lifshitz equation.''

--- We thank the referee for these suggestions. We have expanded the discussion of quantum-classical correspondence to point out that the similarity of the steady states is responsible for the correspondence, and added an appendix with the results of some simulations with a model that interpolates between the integrable classical chain and the non-integrable classical Heisenberg chain.

---

## Round 2 · Referee Report · Anonymous (Referee 2) · 2024-10-10

Strengths

1 - study of a topical experiment/simulation in a NISQ computer 2- unusual result of persistence of ballistic transport in the presence of disorder

Weaknesses

1 - model could be explained in more detail

Report

The authors study both analytically and numerically a recent experiment by Google simulating a boundary driven XXZ spin chain. They focus on the classical, large S, limit and find good agreement with experiment. A very interesting result is their finding of persistence of ballistic transport even in the presence of disorder in the easy plane regime. This stability is because the disorder cannot modify the consistency equation significantly. This is an important point.

Requested changes

1 - the model is not very clear. I only understand what the driving was when I read the caption of Fig. 1. They do discuss it later around Eq. (6), but I suggest that the authors discuss this in more detail and earlier in the text (around Eq. (1)).

2- The discussion about why the ballistic transport is stable to disorder is too short in my opinion. In fact, most of it is in Appendix C. I would suggest expanding this discussion with perhaps an analytical and numerical example to clarify.

Recommendation

Ask for minor revision

  • validity: high
  • significance: high
  • originality: high
  • clarity: high
  • formatting: perfect
  • grammar: perfect

Author:  Adam McRoberts  on 2024-11-28  [id 5004]

(in reply to Report 2 on 2024-10-10)

Response to the Report of Referee 2

''The authors study both analytically and numerically a recent experiment by Google simulating a boundary driven XXZ spin chain. They focus on the classical, large S, limit and find good agreement with experiment. A very interesting result is their finding of persistence of ballistic transport even in the presence of disorder in the easy plane regime. This stability is because the disorder cannot modify the consistency equation significantly. This is an important point.''

--- We thank the referee for their report, and are pleased that they find the results interesting and could recommend publication after minor revisions. We address their specific concerns below.

''1 - the model is not very clear. I only understand what the driving was when I read the caption of Fig. 1. They do discuss it later around Eq. (6), but I suggest that the authors discuss this in more detail and earlier in the text (around Eq. (1)).''

--- We thank the referee for pointing out that this could benefit from some clarification. We have expanded the discussion of the model around Eq.~(1). In particular, we have split the three parts of the timestep into bullet points for easier reading, and explicitly added the equations of motion that govern the bond evolution within each half-timestep.

''2 - The discussion about why the ballistic transport is stable to disorder is too short in my opinion. In fact, most of it is in Appendix [D]. I would suggest expanding this discussion with perhaps an analytical and numerical example to clarify.''

--- We have added a short description to \S4.2 of the ways in which the steady states don't change -- namely that they remain monotonic, the value through the bulk is $z \approx 0$, and the difference $1 - z_{L-1}$ remains non-zero. We have also explicitly remarked that this difference needs to remain non-zero for all realisations of disorder.

---

## Round 3 · Author Response

Dear Editor,

We are writing to resubmit our article, 'Ballistic conductance with and without disorder in a boundary-driven XXZ spin chain', for consideration for publication in SciPost Physics.

We thank you for communicating to us the referee reports, and we thank the referees for their time and assessment of our work.

We are pleased that the referees were broadly positive in their assessment and found the results interesting. We thank them also for their helpful comments and suggestions, which we address in our replies to their reports.

We look forward to your response in due course.

Yours sincerely,

Adam McRoberts and Roderich Moessner

---

## Round 3 · List of Changes

--- We have clarified the details of the model near Eq.~(1), splitting the three parts of the timestep across some bullet points to make this easier to read, and explicitly adding the equations of motion.

--- We have expanded the discussion of the quantum-classical correspondence and the role of integrability in \S3.4, pointing out that it is the similarity of the steady states that is responsible for the correspondence, and pointing to the appendix on integrability.

--- We have added an appendix which reports the results of simulations of a model that interpolates between the integrable classical chain and the non-integrable classical Heisenberg chain, showing that the same asymptotic transport regime is observed in all cases (quantum-integrable, classical-non-integrable, and classical-integrable).

--- We have expanded the discussion of the stability of ballistic transport at the start of \S4.2, describing the stability of the steady-states in more detail, and explicitly remarking that the difference $1 - z_{L-1}$ is finite for all configurations of disorder.

---

## Editorial Decision

published